# Maternal Pre-Pregnancy BMI, Offspring Adiposity in Late Childhood, and Age of Weaning: A Causal Mediation Analysis

**DOI:** 10.3390/nu15132970

**Published:** 2023-06-29

**Authors:** Jie Zhang, Gemma L. Clayton, Stefan Nygaard Hansen, Anja Olsen, Deborah A. Lawlor, Christina C. Dahm

**Affiliations:** 1Department of Public Health, Aarhus University, DK-8000 Aarhus, Denmark; jiezh@ph.au.dk (J.Z.); anja@cancer.dk (A.O.); 2Population Health Sciences, Bristol Medical School, University of Bristol, Bristol BS8 2BN, UK; gemma.clayton@bristol.ac.uk (G.L.C.); d.a.lawlor@bristol.ac.uk (D.A.L.); 3MRC Integrative Epidemiology Unit, Bristol BS8 2BN, UK; 4Danish Cancer Society Research Center, DK-2100 Copenhagen, Denmark

**Keywords:** body mass index, age of weaning, mediation, ALSPAC

## Abstract

Infant feeding practices have been hypothesized to influence offspring’s body mass index (BMI) later in life, and women with overweight or obesity tend to wean their infants earlier than women with healthy BMI. We, therefore, aimed to investigate how much early age of weaning mediated the maternal-offspring adiposity relationship. The study included 4920 mother-child pairs from the Avon Longitudinal Study of Parents and Children birth cohort. G-computation was applied to estimate the natural direct (NDE) and indirect (NIE) effects, via the age of weaning (<3 months, 3 months, >3 months), of maternal pre-pregnancy overweight or obesity on offspring’s BMI and fat mass index. The NDE of maternal overweight or obesity on offspring BMI at 17 years old was 2.63 kg/m^2^ (95% CI: 2.27 to 2.99). The NIE via the age of weaning was 0.02 kg/m^2^ (95% CI: 0.00 to 0.04), corresponding to 0.8% of the total effect. Similar results were observed for the offspring’s fat mass index. The NDE and NIE were similar to the main analyses when we looked at the relationship stratified by breastfeeding status. Our study found a minimal influence of age of weaning on the pathway between maternal and offspring adiposity, indicating the age of weaning may not be a key mediator.

## 1. Introduction

Maternal overweight or obesity is linked to offspring adiposity throughout childhood as well as adulthood [1,2], with various risk factors contributing to the intergenerational transmission of adiposity at different life stages. These pathways include genetic predisposition [3], shared familial lifestyles [4], postnatal nutrition [5], and potentially ‘fetal programming’ [6].

To date, studies aiming to quantify the mechanisms underlying the maternal-offspring adiposity association mainly focus on intrauterine pathways involving gestational weight gain [4] or birth anthropometry [7,8], but results suggest that only a small proportion of the maternal-offspring adiposity association is mediated through these mechanisms [9]. In addition, randomized trials and systematic reviews found little evidence that prenatal lifestyle interventions on women with obesity during pregnancy affect the child’s weight or growth [10,11]. Therefore, identifying potential modifiable mediators between maternal and offspring adiposity to prevent the development of obesity in children and youth remains a challenge. Weaning, defined as an infant’s transition from milk-based feeding to solid foods, is a milestone in child development [12]. Time of weaning may affect the infant’s metabolome, alter energy metabolism, promote different colonization, or even influence behavioral responses to food and eating, all of which could influence an individual’s susceptibility to obesity [13]. Evidence is accumulating that early introduction to solid food is associated with a higher infant growth rate and risk of obesity in childhood or even adolescence [14,15]. Mothers who are overweight or obese might have a predisposition to early weaning due to socio-cultural (i.e., less social support, stress, or depression) or physiological factors (i.e., lactation insufficiency) [16,17]. Therefore, the age of weaning might provide a window of opportunity to intervene in childhood weight development [18].

In this study, we hypothesized that age of weaning was a mediator between maternal pre-pregnancy overweight or obesity and offspring adiposity in late childhood. We used data from a large population-based study and applied causal mediation analysis to quantify whether and how much age of weaning mediates the relationship between maternal and offspring adiposity at age 17 years. Using the G-formula, we estimated the specific contribution of the age of weaning while taking intermediate confounding into account. Furthermore, we stratified the mediation analyses by breastfeeding status to explore the potential modifying effect of breastfeeding.

## 2. Materials and Methods

### 2.1. Study Participants

The study used data from the Avon Longitudinal Study of Parents and Children (ALSPAC). All pregnant women in the former Avon Health Authority with an expected delivery date between 1 April 1991 and 31 December 1992 were eligible for the study. The original cohort consisted of 15,443 infants. Of those children, 10,266 were excluded due to death before 1 year of age or twin or triple births (*n* = 764), missing maternal pre-pregnancy anthropometrics (*n* = 3312), or lacking self-reported or clinic-measured anthropometrics at age 17 years (*n* = 6190). Of eligible participants, 257 participants were excluded due to missing weaning information. As a result, we included 4920 mother-offspring pairs in the analyses (Figure 1). Further information on the ALSPAC cohort is available on the ALSPAC website (http://www.bristol.ac.uk/alspac; accessed on 8 June 2023) and elsewhere [19,20]. Ethical approval for the study was obtained from the ALSPAC Ethics and Law Committee and the Local Research Ethics Committees (http://www.bristol.ac.uk/alspac/researchers/research-ethics/ (accessed on 8 June 2023), ALSPAC project ID B3425, approved 6 August 2020).

Informed consent for the use of data collected via questionnaires and clinics was obtained from participants following the recommendations of the ALSPAC Ethics and Law Committee at the time.

### 2.2. Exposure: Maternal Pre-Pregnancy BMI

Mothers’ pre-pregnancy weight and height were self-reported on questionnaires completed at 12 weeks of gestation. The mothers also reported their partners’ weight and height on a questionnaire sent to them when the mothers were approximately 18 weeks pregnant. Body mass index (BMI) was calculated as weight (kg) divided by height (m) squared. The primary exposure was maternal pre-pregnancy BMI categorized as ‘Normal weight’ (BMI < 25 kg/m^2^) and ‘Overweight or obese’ (BMI ≥ 25 kg/m^2^).

### 2.3. Outcome: Offspring Adiposity

The main outcome was offspring BMI at the age of approximately 17 years. During adolescence (mean age 17.7; SD = 0.5 years), children attended follow-up clinics where weight and height were measured with the participant in light clothing and without shoes. Weight was measured to the nearest 0.1 kg with Tanita scales and height to the nearest 0.1 cm using a Harpenden stadiometer. The secondary outcome was the fat mass index (FMI). Fat mass (kg) and lean mass (kg) were assessed by a Lunar Prodigy dual-energy X-ray absorptiometry (DXA) scanner (GE Medical Systems Lunar, Madison, WI, USA) [21]. The scans were visually inspected and realigned where necessary. Once complete, the tester examined the scan to ensure its quality and, if necessary, repeated the scan. FMI (kg/m^2^) was calculated as fat mass (in kg) divided by height (m) squared.

### 2.4. Mediator: Age of Weaning

On a questionnaire completed at 6 months postpartum, mothers reported at which age their child started solid food (in months). Because a common length of maternity leave for this cohort was 12 weeks, and as a return to work may change infant feeding practices, age of weaning was categorized into: (i) introducing solid food before 3 months, (ii) at 3 months, and (iii) after 3 months (which included those reporting not starting yet).

### 2.5. Covariates

Mothers’ smoking status and alcohol consumption during pregnancy were reported in the mothers’ questionnaires at 18 weeks of gestation. Maternal education level and parity were assessed by questionnaire at 32 weeks of gestation. Breastfeeding and formula feeding were assessed in multiple post-natal questionnaires administered to the mother; breastfeeding duration was defined as (i) never breastfeeding; (ii) stopped breastfeeding at 6 months (1 day to 6 months of duration); (iii) still breastfeeding at 6 months (>6.01 months). Formula feeding was dichotomized to ever or never formula fed. Gestational age at birth was estimated from clinical records. Birth weight was measured by trained study staff where the mother gave permission, or if these data were missing, the values were extracted from the medical records by trained study staff. Information on the sex of the child was obtained from obstetric records.

### 2.6. Assumed Causal Relationships

Figure 2 presents a directed acyclic graph (DAG) of the hypothesized effect of maternal pre-pregnancy overweight or obesity on offspring adiposity. The direct effect represents the amount of the total effect of maternal pre-pregnancy overweight/obesity on a child’s adiposity that does not operate via the age of weaning (pathway b). Therefore, the direct effect corresponds to the difference in offspring’s adiposity that would be observed if we compared mothers with overweight or obese to mothers with normal weight but at the same time could fix the age of weaning to what it would be if all mothers were normal weight. The indirect effect represents the portion of the total effect of maternal pre-pregnancy overweight/obesity on a child’s adiposity that operates via the age of weaning (pathway a + c). The sum of these two effects represents the total effect of the exposure on the outcome (pathways a + c and b).

Maternal age at delivery, maternal highest educational attainment, maternal smoking and drinking status during pregnancy, and child sex were identified as possible confounders for all three pathways and were controlled for in all adjusted analyses. Birthweight was treated as an intermediate confounder as it might be affected by maternal weight status, but also confound the association between age of weaning and offspring adiposity. Breastfeeding was not treated as a confounding variable in the analyses because it is unlikely that the observed association between breastfeeding and overweight is causal [22,23].

### 2.7. Statistical Analyses

Statistical analyses were conducted in Stata/BE version 17. We performed multinomial logistic and linear regressions to describe associations between (1) maternal pre-pregnancy weight status (overweight/obesity vs normal) and age of weaning (<3 m, =3 m, >3 m), and (2) age of weaning and offspring BMI at 17 years old (continuous).

We estimated direct and indirect effects under the following assumptions: (a) no unmeasured confounding for the exposure-outcome relationship, (b) no unmeasured confounding for the mediator–outcome relationship, (c) no unmeasured confounding for the exposure-mediator relationship, and (d) no effects of the exposure that confounds the mediator–outcome relationship. The total causal effect (TCE) between maternal overweight/obesity and offspring’s BMI is decomposed into the natural direct effect (NDE, i.e., effect of the exposure through other mechanisms than the mediator), and the natural indirect effect (NIE, i.e., the effect of the exposure acting through the mediator). Causal mediation analyses were conducted using the g-formula package in Stata [24,25], which used Monte Carlo simulations with 1000 repetitions for bootstrapping. In addition, we performed a sequential mediation analysis [25] to estimate mediation via both birth weight and weaning age, based on the assumption of a causal ordering of the mediators. The same analyses were conducted for the secondary outcome, FMI at 17 years old.

We further examined whether breastfeeding status modified the relationship between maternal pre-pregnancy overweight/obesity and offspring BMI at age 17 by conducting conditional mediation analysis [26]. We conducted this by adding breastfeeding status as a moderator of the indirect effect at three levels: (i) Never breastfeeding; (ii) stopped breastfeeding at 6 months; (iii) still breastfeeding at 6 months or later.

### 2.8. Sensitivity Analyses

We performed a sensitivity analysis to assess and account for violations of the uncontrolled confounding assumption. A binary unmeasured confounding variable U was assumed to be a common cause of weaning age and offspring BMI (e.g., social and financial support, health knowledge, maternal psychological stress, or infant sleep) [27,28] (Appendix A).

We assumed that offspring with the presence of U had on average 0.2 units higher BMI (kg/m^2^). We also considered a simplified assumption that the prevalence of U was the same among different maternal weight groups. We evaluated the impact of an unmeasured mediator–outcome confounding in 3 settings: (i) mild confounding, individuals with mothers with overweight or obesity had 0.2 higher probability of having U present conditional on weaning age; (ii) moderate confounding, individuals with mothers with overweight or obesity had 0.3 higher probability of having U present conditional on weaning age; and (iii) strong confounding, individuals with mothers with overweight or obesity had 0.4 higher probability of having U present conditional on weaning age. The sensitivity analyses were conducted using the bias formula proposed by VanderWeele [29] (details available in the Appendix A).

## 3. Results

### 3.1. Characteristics of Participants

A total of 4920 mother-offspring pairs were included in the analyses, with valid data for maternal pre-pregnancy weight status, age of weaning, and offspring’s BMI measurement at age 17 years old. Table 1 shows the characteristics of the study population based on the age of weaning. The mean age of weaning in the sample of children in the final analysis was 3.2 months (SD = 0.9), with the peak timing of introduction of solids at 3 months. Infants born to mothers who were overweight/obese before pregnancy were more likely to be weaned before the age of 3 months (16%, *n* = 157) compared to those born to mothers of normal weight (12%, *n* = 475). Younger mothers, who were with lower educational levels, or who smoked before or during pregnancy were more likely to introduce solid food before 3 months. Male infants or those with a higher birth weight were also more likely to be weaned before 3 months. Infants who were never breastfed, or were formula-fed were more likely to start eating solid meals prior to 3 months (Table 1). Compared to the full cohort, children excluded from the analyses were more likely to have mothers with higher BMI, who smoked during pregnancy and who had lower levels of education (Appendix A). Missing data were also more prevalent in male children, or children who were given formula instead of breastmilk, and introduced to solid meals before the age of three months.

### 3.2. Linear and Multinomial Logistic Regression Results

Results from linear regression models are provided in Appendix A. At age 17 years, offspring BMI was 2.7 kg/m^2^ (95% CI: 2.4 to 3.0) higher for those born with mothers with overweight/obesity compared with those whose mothers had normal weight, with adjustment for maternal education, age, smoking, drinking, and offspring sex. Earlier age of weaning was related to offspring BMI at age 17 years. Offspring BMI was 0.6 kg/m^2^ (95% CI: 0.2 to 1.0) higher at age 17 for infants who were introduced to solid food prior to 3 months compared to those introduced after 3 months (Appendix A). Multinomial logistic regression models show that for women with overweight/obesity relative to those with normal weight, the ratio of relative risks (RRs) for weaning before 3 months versus over 3 months was 1.40 (95% CI: 1.10 to 1.78), indicating a 40% greater RR for early weaning among women with overweight/obesity. The corresponding ratio of RRs for weaning at 3 months was 1.13 (95% CI: 0.95 to 1.34) (Appendix A).

### 3.3. Causal Mediation Results

Results from the causal mediation analyses are shown in Table 2, including the TCE, NDE, and NIE for the effect of maternal pre-pregnancy overweight/obesity status on offspring BMI at age 17 years. The NDE shows that offspring BMI increased on average by 2.6 kg/m^2^ (95% CI: 2.3 to 3.0) when altering the levels of maternal pre-pregnancy BMI from normal weight to overweight/obesity while controlling for the age of weaning (i.e., setting the age of weaning to levels naturally observed for normal-weight mothers). The NIE was very small (0.02 kg/m^2^, 95% CI: 0.00 to 0.04) when comparing the age of weaning at the values that would naturally occur for normal-weight mothers versus mothers with overweight/obesity while controlling maternal pre-pregnancy BMI categories at the overweight/obesity level. The proportion of the total effect mediated by age of weaning was 0.8% (95% CI: 0.1% to 1.6%).

Similarly, the relationship between maternal pre-pregnancy overweight/obesity status and offspring FMI was largely explained by the NDE, while the NIE of the age of weaning was negligible (Appendix A). The NDE and NIE were similar to the main analyses when we looked at the relationship in women who never breastfed, those who stopped breastfeeding before 6 months, and those who were still breastfeeding at 6 months (Table 3).

When looking at the combined mediating effect of birth weight and age of weaning, the NIE was 0.2 kg/m^2^ (95% CI: 0.1 to 0.2), which accounts for 5% of the combined total effect (Table 4).

### 3.4. Sensitivity Analyses

The sensitivity analyses found that even under the worst confounding scenario (strong unmeasured mediator–outcome confounding), the majority of the effect was still due to the direct effect of maternal weight status. The overall conclusion thus remained similar (Appendix A).

## 4. Discussion

Using data from the ALSPAC study, our study assessed the path-specific mediation effect of age of weaning between maternal pre-pregnancy overweight/obesity status and offspring’s adiposity at age 17. Contrary to our hypotheses, only 0.8% of the total effect was mediated by age of weaning, indicating that age of weaning might not be a key mediator on the causal pathway between maternal and offspring adiposity in early adulthood. Relative to the natural indirect effect through the age of weaning, the magnitude of the natural direct effect of maternal pre-pregnancy weight status was much stronger. This finding indicates that the mother-offspring adiposity association at age 17 was mainly through the direct effect of maternal pre-pregnancy overweight/obesity or through other pathways not investigated here.

The link between maternal pre-pregnancy overweight/obesity and adiposity in their children has been consistently highlighted in the literature, yet the underlying mechanisms of this relationship are not fully understood [30]. Accumulating evidence suggests that the time of introducing solid food is critical in establishing long-term dietary habits and is a significant risk factor for the development of childhood obesity, independent of the weaning food consumed [31]. Early age of weaning may be associated with higher adiposity risk due to epigenetic changes in metabolic programming [32] or hormone levels related to hunger or food cravings, which further raise body fat levels [33]. Mothers with overweight or obesity before pregnancy are more likely to introduce solid food to infants earlier than mothers with normal weight [15,32,34,35,36,37]. The reasons behind the decision might be due to physiological factors like lactation insufficiency or lack of social support in terms of feeding guidelines [16,17] but appear to be independent of current maternity leave policies [38]. Nevertheless, our results indicate that earlier age of weaning among children of mothers with obesity or overweight is not an important driver of the strong and consistent association between maternal and offspring BMI in late childhood.

To our knowledge, no previous studies have explored the path-specific effect of age of weaning between maternal pre-pregnancy overweight/obesity and child adiposity in late childhood. Consistent with our findings, several mediation studies investigating other potential mediators have reported that the direct effect of pre-pregnancy maternal overweight/obesity on child adiposity risk was stronger than indirect effects via mediators such as birth anthropometry [7,9,39,40]. For example, a US cohort study included birth anthropometric measures (weight, length, head circumference, ponderal index, small-for-gestational-age or large-for-gestational-age) as mediators between maternal pre-pregnancy weight status and child’s overweight/obesity in 3950 mother-child pairs, but found only a small portion was mediated by birth weight and head circumference7. Similarly, an Australian longitudinal study showed that pre-pregnancy BMI had a significant natural direct effect on children’s BMI z score (beta = 0.14, 95% CI: 0.06 to 0.22 per kg/m^2^), while the indirect effect of birth weight was not significant [9]. The consistent and strong effect of maternal pre-pregnancy BMI on childhood adiposity may arise from shared genetics or from postnatal behaviors not considered here, like food choice [2,41]. Future studies should explore other mediators to disentangle the causal relationship and formulate effective prevention strategies.

Compared to previous mediation studies that only examined offspring BMI during childhood, our study followed children until the age of 17 to gain a better understanding of the risk factors for obesity in late childhood and early adulthood. The long-term follow-up design from maternal pre-pregnancy to early adulthood in the next generation ensures the correct temporal ordering of exposure, mediator, and outcome. Another strength of the present study is the counterfactual approach, allowing us to assess the path-specific effect of age of weaning and to control for confounding factors along each step of the exposure-mediator–outcome path. We used the G-formula and conducted a series of sensitivity analyses to test the robustness of the results. Similar results were observed in both main and secondary analyses.

There are some limitations in the study. First, the mediation analyses rest on strong assumptions of no unmeasured confounding [42], which in practice cannot be ruled out. For instance, the shared genetic predisposition to obesity could not be adjusted for here [43]. Residual confounding may lead to an overestimation of the total and direct effects in our study [44]. However, our sensitivity analyses testing the robustness of the results suggest that the main conclusions would not be altered, even under the worst-case scenario. Another limitation is that our results could be affected by measurement error, particularly in maternal reports of her pregnancy weight and height. However, this has been shown to correlate highly with measurements from the first antenatal clinical visit [45]. Therefore, bias on the direct or indirect effects [46] due to exposure measurement error should not be substantial. Mothers were also asked to report the month ‘when babies start solid food’ on the 6-month postpartum questionnaire, which may be particularly erroneously recalled for babies that were weaned early [47]. However, the date of weaning was reported prior to future measurements of the child’s anthropometrics, thus the measurement error would tend to be non-differential and thus bias the indirect effect towards the null [42]. It is also possible that specific types of food are important mediators, however, we could not delve deeper into the details of what solid food was introduced, nor the composition or quality of the supplementary food as our research question was to investigate the timing of introduction of solid food [48].

It is worth noting the weaning practices reported in the present study reflected social and structural practices in the early 1990s. Current guidance from the World Health Organization (WHO) consistently recommends complementary feedings should begin no earlier than 6 months [49]. Despite this advice, successive infant feeding surveys have shown that the majority of UK children continue to be weaned before the age of 5 months [50]. Though we did not find the age of weaning as a significant contributor to the causal pathway between maternal and offspring adiposity in late childhood, there is still evidence showing very early age of weaning per se is associated with a higher risk of later adiposity [12,14,15,51,52]. Within that context, it has been suggested age of weaning might be a possible avenue for the prevention of the development of obesity [14,31].

## 5. Conclusions

Under causal assumptions, this study observed that maternal pre-pregnancy overweight/obesity and early age of weaning were related to higher offspring BMI in late childhood. However, there was little evidence suggesting that age of weaning mediates the causal pathway between maternal and offspring adiposity in late childhood. Future studies should explore other modifiable factors that could mediate the intergenerational transmission of adiposity.

## Figures and Tables

**Figure 1 nutrients-15-02970-f001:**
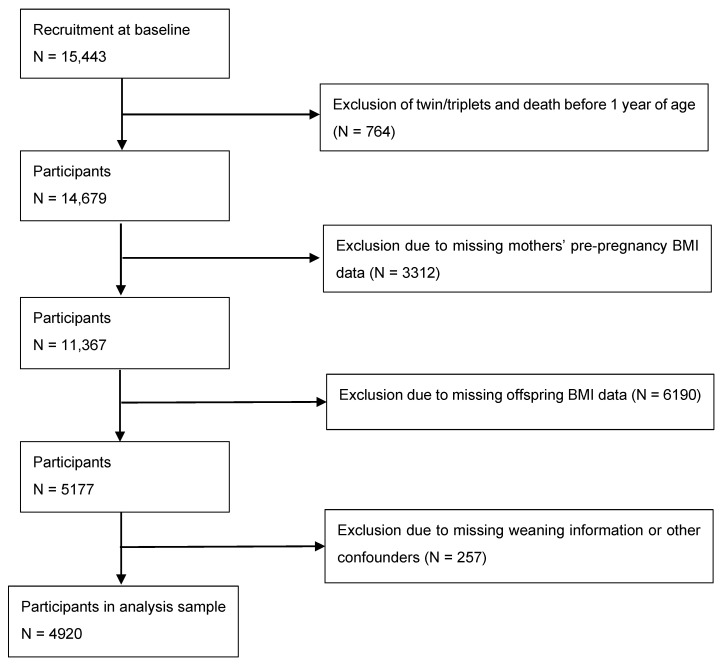
Flow chart of Avon Longitudinal Study of Parents and Children participants in the study.

**Figure 2 nutrients-15-02970-f002:**
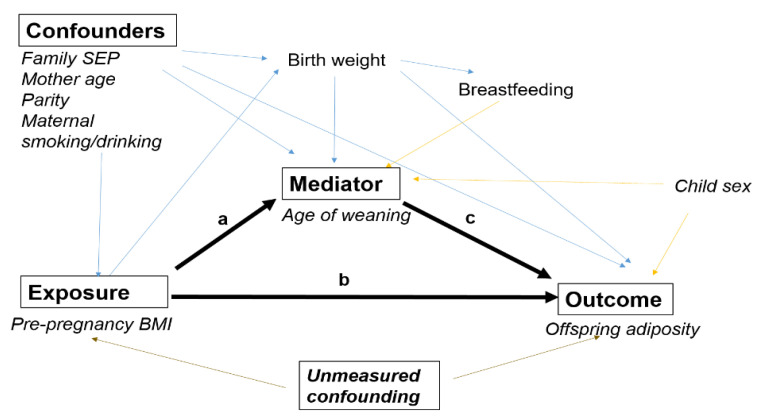
Directed acyclic graph showing the hypothesized relationship between maternal pre-pregnancy body mass index and the child’s adiposity. The indirect effect operates via path a + c, while the direct effect operates via path b. The total effect is the sum of paths a + c and b.

**Table 1 nutrients-15-02970-t001:** Characteristics of participants by age of weaning.

		Age of Weaning						
		<3 Months		=3 Months		>3 Months		Total
**Categorical variable**		*n*	%	*n*	%	*n*	%	*n*
		632	12.85	2839	57.7	1449	29.45	4920
Maternal pre-pregnancy BMI	Normal	475	12.01	2283	57.71	1198	30.28	3956
	OWOB	157	16.29	556	57.68	251	26.04	964
Maternal highest educational attainment	CSE	110	20.37	314	58.15	116	21.48	540
	Vocational	51	14.45	209	59.21	93	26.35	353
	O level	242	14.4	1011	60.18	427	25.42	1680
	A level	163	11.99	769	56.54	428	31.47	1360
	Degree	55	5.99	501	54.58	362	39.43	918
Maternal smoking habits during pregnancy	Never	312	11.03	1608	56.86	908	32.11	2828
	Stopped smoking during pregnancy	206	14.8	809	58.12	377	27.08	1392
	Always Smoking	105	16.67	378	60	147	23.33	630
Maternal drinking habits during pregnancy	Never	291	13.8	1187	56.31	630	29.89	2108
	<1 glass per week	238	11.8	1185	58.75	594	29.45	2017
	≥1 glass per week	93	12.4	440	58.67	217	28.93	750
Child sex	Male	330	14.99	1284	58.31	588	26.7	2202
	Female	302	11.11	1555	57.21	861	31.68	2718
Ever breastfed	Never breastfed	163	20.58	467	58.96	162	20.45	792
	Stopped breastfeeding at 6 months	341	14.74	1405	60.72	568	24.55	2314
	Still breastfeeding at 6 months	125	6.99	950	53.16	712	39.84	1787
Formula-fed	Never	82	8.06	486	47.74	450	44.2	1018
	Ever	550	14.1	2353	60.3	999	25.6	3902
**Continuous variable**		**Mean**	**SD**	**Mean**	**SD**	**Mean**	**SD**	**Mean (SD)**
Maternal age		27.8	4.6	29.2	4.4	30.5	4.4	29.4 (4.5)
*n*		612		2763		1416		4791
Maternal BMI (kg/m^2^)		23.5	4.3	22.8	3.5	22.6	3.5	22.9 (3.6)
*n*		632		2839		1449		4920
Paternal BMI (kg/m^2^)		25.2	3.3	25.1	3.2	24.9	3.1	25.0 (3.2)
*n*		470		2160		1104		3734
Child BMI at age 17 years (kg/m^2^)		23.4	4.8	22.6	3.9	22.4	3.9	22.6 (4.1)
*n*		632		2839		1449		4920
Child age (years)		17.6	0.5	17.7	0.5	17.7	0.5	17.7 (0.5)
*n*		632		2839		1449		4920
Child birthweight (kg)		3.5	0.6	3.44	0.5	3.4	0.6	3.4 (0.5)
*n*		625		2798		1435		4858
Child gestational age (weeks)		39.7	1.6	39.6	1.6	39.2	2.1	39.5 (1.8)
*n*		632		2839		1449		4920

BMI, body mass index; OWOB, overweight or obesity; SD, standard deviation.

**Table 2 nutrients-15-02970-t002:** Causal mediation effects of age of weaning on the association between maternal and offspring BMI at 17 years.

	G-Computation Estimate (MD)	95% CI	*p* Value
TCE	2.65	2.30 to 3.01	0.00
NDE	2.63	2.27 to 2.99	0.00
NIE	0.02	0.003 to 0.042	0.03
PM	0.008	0.0009 to 0.0159	0.03
CDE ^a^	2.63	2.27 to 2.99	0.00

NIE is expressed as the expected mean difference in offspring’s BMI when the age of weaning takes the value observed in individuals with mothers who were overweight or obese compared to those with normal weight. PM is estimated as NIE/TCE. Bootstrapping was used to obtain the CI. The mediator model was a multinomial logistic regression and the outcome model was a linear regression. TCE, total causal effect; NDE, natural direct effect; NIE, natural indirect effect; CDE, controlled direct effect; PM, proportion mediated; BMI, body mass index; CI, confidence interval; MD, mean difference. ^a^ CDE, controlled at the age of weaning after 3 months.

**Table 3 nutrients-15-02970-t003:** Causal mediation effects stratified by breastfeeding status.

Breastfeeding	NDE ^a^	95% CI	NIE ^b^	95% CI
Never breastfed	2.62	1.79 to 3.47	0.006	−0.029 to 0.041
Stop breastfeeding at 6 months	2.76	2.25 to 3.27	0.015	−0.013 to 0.043
Still breastfeeding at 6 months	2.37	1.74 to 3.00	0.027	−0.017 to 0.070
Total	2.63	2.27 to 2.99	0.020	0.003 to 0.042

NDE, natural direct effect; NIE, natural indirect effect; CI, confidence interval. ^a^ NDE, natural direct effect of maternal weight status on offspring’s BMI at 17 years. ^b^ NIE, natural indirect effect of maternal weight status on offspring’s BMI at 17 years via the age of weaning.

**Table 4 nutrients-15-02970-t004:** Sequential mediation analysis of the indirect effect through birth weight and age of weaning.

Mediators	G-Computation Estimate (MD)	95% CI	*p* Value
Both birthweight and age of weaning	0.15	0.07 to 0.24	0.00
Birth weight	0.10	0.06 to 0.14	0.00
Age of weaning	0.05	−0.02 to 0.13	0.17

MD, mean difference; CI, confidence interval.

## Data Availability

Data requests are made to the ALSPAC executive using the form available from the study website: http://www.bristol.ac.uk/alspac/researchers/access/ (accessed on 8 June 2023). Guidance for researchers and collaborators, the study protocol, and the data collection schedule are all available via the website. All requests are carefully considered and accepted where possible. The manuscript has been approved by the ALSPAC executive.

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
