# Peer review of "Maternal Pre-Pregnancy BMI, Offspring Adiposity in Late Childhood, and Age of Weaning: A Causal Mediation Analysis"

_nutrients, 2023, doi:10.3390/nu15132970_

Round 1
Reviewer 1 Report
This paper aims to investigate the role of early weaning on the association between maternal and offspring adiposity. The hypotheses of the paper are clearly stated, the approach used is well explained and the sensitivity analyses performed, substantiate the robustness of the results. The limitations of the study and in particular of causal mediation analysis are well discussed.
I only have one question about this paper. As the magnitude of the effects of M on Y can vary greatly depending on the chosen interval, I am wondering how the effect is attenuating over time. I know that there are many follow-up waves in ALSPAC so you should be able to eventually have a look at previous time point. I won't be surprised that the mediation effect may be stronger at earlier time point.
I will probably be less firm when concluding that age of weaning is not a key mediator as it may depend on the time interval considered. It is not a stationary process.
Author Response
This paper aims to investigate the role of early weaning on the association between maternal and offspring adiposity. The hypotheses of the paper are clearly stated, the approach used is well explained and the sensitivity analyses performed, substantiate the robustness of the results. The limitations of the study and in particular of causal mediation analysis are well discussed.
I only have one question about this paper. As the magnitude of the effects of M on Y can vary greatly depending on the chosen interval, I am wondering how the effect is attenuating over time. I know that there are many follow-up waves in ALSPAC so you should be able to eventually have a look at previous time point. I won't be surprised that the mediation effect may be stronger at earlier time point.
I will probably be less firm when concluding that age of weaning is not a key mediator as it may depend on the time interval considered. It is not a stationary process.
Response: Thank you for your valuable suggestion. We acknowledge and agree that the relationship between age of weaning and offspring adiposity may depend on the time interval of the study design. The scope of the paper was to assess whether the association observed between maternal pre-pregnancy BMI and offspring BMI/fat mass at late childhood was due to mediation through age of weaning. To the best of our knowledge, no previous study has examined the long-term impact of maternal pre-pregnancy BMI over such a long follow-up period. In response to your comment, we have clarified in the manuscript that age of weaning is not a key mediator of maternal-offspring adiposity association in late childhood. Please see changes in Line 27-28, Line 52, and Line 159, Line258, Line 260, Line 263, Line 280, Line 283, Line 300, Line 333, Line339, and Line 341.
Reviewer 2 Report
I am honored to review an ALSPAC study because it is a cohort from which we have learned much. I can only find three areas for improvement in the article.
1. I think it is very good that the limitations include the fact that the stratification of weaning time (<3m, 3m, >3m) is not in line with the current trend (>6m). But I also think this should be written in the abstract or the title to not confuse those who do not read the whole article.
2. The authors write that breastfeeding has not been treated as a confounder because the causal relationship between breastfeeding and overweight is doubtful. I have two objections to this decision. On the one hand, almost all causal associations in pediatrics are dubious. On the other hand, because the relationship is doubtful, there is no argument for deciding that breastfeeding does not deserve to be treated as a confounder.
3. I would like to add that weaning is an unhelpful term because it covers too many foods. It should not have the same consequences for a young infant to start eating fruit or vegetables as it does sugary flour with cow's milk.
Author Response
I am honored to review an ALSPAC study because it is a cohort from which we have learned much. I can only find three areas for improvement in the article.
1. I think it is very good that the limitations include the fact that the stratification of weaning time (<3m, 3m, >3m) is not in line with the current trend (>6m). But I also think this should be written in the abstract or the title to not confuse those who do not read the whole article.
Response: We appreciate the feedback. We have added the stratification of weaning time in the abstract, please see Line 16-17.
The authors write that breastfeeding has not been treated as a confounder because the causal relationship between breastfeeding and overweight is doubtful. I have two objections to this decision. On the one hand, almost all causal associations in pediatrics are dubious. On the other hand, because the relationship is doubtful, there is no argument for deciding that breastfeeding does not deserve to be treated as a confounder.
Response: We have rephrased the sentence to ‘Breastfeeding was not treated as a confounding variable in the analyses because it is unlikely that the observed association between breastfeeding and overweight is causal’.
Furthermore, we would like to emphasize that the primary mediator of interest in this manuscript was the age of weaning, and breastfeeding was treated as moderator of the indirect effect between age of weaning and offspring adiposity. Thus we conducted subgroup analyses to investigate the indirect effect at three different breastfeeding levels.
I would like to add that weaning is an unhelpful term because it covers too many foods. It should not have the same consequences for a young infant to start eating fruit or vegetables as it does sugary flour with cow's milk.
Response: We appreciate your comment regarding the term "weaning" and its broad implications. We agree that the term encompasses a wide range of foods, and the consequences of introducing different types of foods during this period can vary significantly.
In our study, we utilized the term "weaning" to refer to the overall transition from 100% milk-based diet to a diet including solid foods. We have clarified this in Line 325-326.